# Adversarial Self-Defense for Cycle-Consistent GANs

Dina Bashkirova [1], Ben Usman[1], and Kate Saenko [1,2]

[1]Boston University
[2]MIT-IBM Watson AI Lab
{dbash,usmn,saenko}@bu.edu

## Abstract

The goal of unsupervised image-to-image translation is to map images from one domain to another without the ground truth correspondence between the two domains. State-of-art methods learn the correspondence using large numbers of unpaired examples from both domains and are based on generative adversarial networks. In order to preserve the semantics of the input image, the adversarial objective is usually combined with a cycle-consistency loss that penalizes incorrect reconstruction of the input image from the translated one. However, if the target mapping is many-to-one, e.g. aerial photos to maps, such a restriction forces the generator to hide information in low-amplitude structured noise that is undetectable by human eye or by the discriminator. In this paper, we show how such self-attacking behavior of unsupervised translation methods affects their performance and provide two defense techniques. We perform a quantitative evaluation of the proposed techniques and show that making the translation model more robust to the self-adversarial attack increases its generation quality and reconstruction reliability and makes the model less sensitive to low-amplitude perturbations. Our project page can be found at `ai.bu.edu/selfadv/`.

## 1  Introduction

Generative adversarial networks (GANs) [7] have enabled many recent breakthroughs in image generation, such as being able to change visual attributes like hair color or gender in an impressively realistic way, and even generate highly realistic-looking faces of people that do not exist [13, 31, 14]. Conditional GANs designed for unsupervised image-to-image translation can map images from one domain to another without pairwise correspondence and ground truth labels, and are widely used for solving such tasks as semantic segmentation, colorization, style transfer, and quality enhancement of images [34, 10, 19, 3, 11, 35, 4] and videos [2, 1]. These models learn the cross-domain mapping by ensuring that the translated image both looks like a true representative of the target domain, and also preserves the semantics of the input image, e.g. the shape and position of objects, overall layout etc. Semantic preservation is usually achieved by enforcing cycle-consistency [34], *i.e.* a small error between the source image and its reverse reconstruction from the translated target image.

Despite the success of cycle-consistent GANs, they have a major flaw. The reconstruction loss forces the generator network to hide the information necessary to faithfully reconstruct the input image inside tiny perturbations of the translated image [5]. The problem is particularly acute in many-to-one mappings, such as photos to semantic labels, where the model must reconstruct textures and colors lost during translation to the target domain. For example, Figure 1's top row shows that even when the car is mapped incorrectly to semantic labels of building (gray) and tree (green), CycleGAN is still able to "cheat" and perfectly reconstruct the original car from hidden information. It also reconstructs road textures lost in the semantic map. This behavior is essentially an adversarial attack that the model is performing on itself, so we call it a *self-adversarial attack*.

| | input | ground truth | translation | noisy translation | reconstruction | noisy reconstruction |
|---|---|---|---|---|---|---|

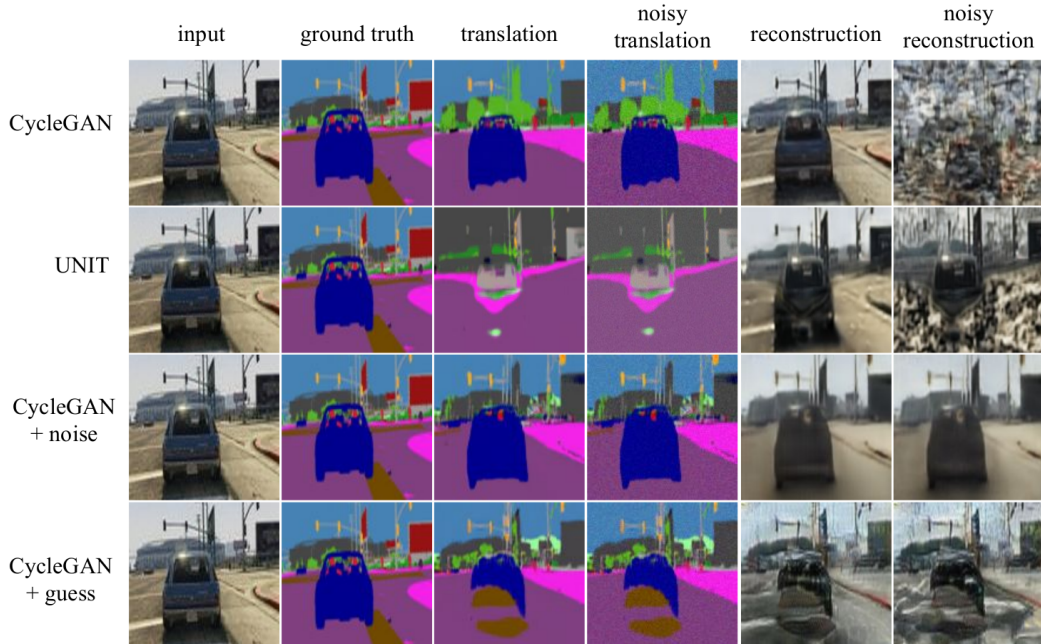

Figure 1: Results of translation of GTA [26] frames to semantic segmentation maps using CycleGAN, UNIT and CycleGAN with our two proposed defense methods, additive noise and guess loss. The last column shows the reconstruction of the input image when high-frequency noise (Gaussian noise with mean 0 and standard deviation $0.08 \sim 10$ intensity levels out of 256) is added to the output map. Ideally, if the reconstruction is "honest" and relies solely on the visual features of the input, the reconstruction quality should not be greater than that of the translation. The results of all three translation methods (CycleGAN, UNIT and MUNIT) show that the reconstruction is almost perfect regardless of the translation accuracy. Furthermore, the reconstruction of the input image is highly sensitive to low-amplitude random noise added to the translation. Both of the proposed self-adversarial defense techniques (Section 4) make the CycleGAN model more robust to the random noise and make it rely more on the translation result rather than the adversarial structured noise as in the original CycleGAN and UNIT. More translation examples can be found in the Section 3 of supplementary material. *Best viewed in color.*

In this paper, we extend the analysis of self-adversarial attacks provided in [5] and show that the problem is present in recent state-of-art methods that incorporate cycle consistency. We provide two defense mechanisms against the attack that resemble the adversarial training technique widely used to increase robustness of deep neural networks to adversarial attacks [9, 16, 32]. We also introduce quantitative evaluation metrics for translation quality and reconstruction "honesty" that help to detect self-adversarial attacks and provide a better understanding of the learned cross-domain mapping. We show that due to the presence of hidden embeddings, state of the art translation methods are highly sensitive to high-frequency perturbations as illustrated in Figure 1. In contrast, our defense methods substantially decrease the amount of self-adversarial structured noise and thus make the mapping more reliant on the input image, which results in more interpretable translation and reconstruction and increased translation quality. Importantly, robustifying the model against the self-adversarial attack makes it also less susceptible to the high-frequency perturbations which make it less likely to converge to a non-optimal solution.

## 2 Related Work

Unsupervised image-to-image translation is one of the tasks of domain adaptation that received a lot of attention in recent years. Current state-of-art methods [34, 20, 11, 15, 4, 10] solve this task using generative adversarial networks [8] that usually consist of a pair of generator and discriminator networks that are trained in a min-max fashion to generate realistic images from the target domain and correctly classify real and fake images respectively.

The goal of image-to-image translation methods is to map the image from one domain to another in such way that the output image both looks like a real representative of the target domain and contains

the semantics of the input image. In the supervised setting, the semantic consistency is enforced by the ground truth labels or pairwise correspondence. In case when there is no supervision, however, there is no such ground truth guidance, so using regular GAN results in often realistic-looking but unreliable translations. In order to overcome this problem, current state-of-art unsupervised translation methods incorporate cycle-consistency loss first introduced in [34] that forces the model to learn such mapping from which it is possible to reconstruct the input image.

Recently, various methods have been developed for unimodal (CycleGAN [34], UNIT [20], CoGAN [21] etc.) and multimodal (MUNIT [11], StarGAN [4], BicycleGAN [35]) image-to-image translation. In this paper, we explore the problem of self-adversarial attacks in three of them: CycleGAN, UNIT and MUNIT. **CycleGAN** is a unimodal translation method that consists of two domain discriminators and two generator networks; the generators are trained to produce realistic images from the corresponding domains, while the discriminators aim to distinguish in-domain real images from the generated ones. The generator-discriminator pairs are trained in a min-max fashion both to produce realistic images and to satisfy the cycle-consistency property. The main idea behind **UNIT** is that both domains share some common semantics, and thus can be encoded to the shared latent space. It consists of two encoder-decoder pairs that map images to the latent space and back; the cross-domain translation is then performed by encoding the image from the source domain to the latent space and decoding it with the decoder for the target domain. **MUNIT** is a multimodal extension of UNIT that performs disentanglement of domain-specific (style space) and domain-agnostic (content space) features. While the original MUNIT does not use the explicit cycle-consistency loss, we found that cycle-consistency penalty significantly increases the quality of translation and helps the model to learn more reliable content disentanglement (see Figure 2). Thus, we used the MUNIT with cycle-consistency loss in our experiments.

As illustrated in Figure 2, adding cycle-consistency loss indeed helps to disentangle domain-agnostic information and enhance the translation quality and reliability. However, such pixelwise penalty was shown [5] to force the generator to hide the domain-specific information that cannot be explicitly reconstructed from the translated image (i.e., shadows or color of the buildings from maps in maps-to-photos example) in such way that it cannot be detected by the discriminator.

It has been known that deep neural networks [17], while providing higher accuracy in the majority of machine learning problems, are highly susceptible to the adversarial attacks [24, 29, 16, 23]. There exist multiple defense techniques that make neural networks more robust to the adversarial examples, such as adding adversarial examples to the training set or adversarial training [24, 22], distillation [25], ensemble adversarial training [30], denoising [18] and many more. Moreover, [33] have shown that defending the discriminator in a GAN setting increases the generation quality and prevents the model from converging to a non-optimal solution. However, most adversarial defense techniques are developed for the classification task and are very hard to adapt to the generative setting.

## 3 Self-Adversarial Attack in Cyclic Models

Suppose we are given a number of samples from two image domains $x \sim p_A$ and $y \sim p_B$. The goal is to learn two mappings $G : x \sim p_A \to y \sim p_B$ and $F : y \sim p_B \to x \sim p_A$. In order to learn the distributions $p_A$ and $p_B$, two discriminators $D_A$ and $D_B$ are trained to classify whether the input image is a true representative of the corresponding domain or generated by $G$ or $F$ accordingly. The cross-distribution mapping is learned using the cycle-consistency property in form of a loss based on the pixelwise distance between the input image and its reconstruction. Usually, the cycle-consistency loss can be described as following:

$$\mathcal{L}_{rec} = \|F(G(x)) - x\|_1 \qquad (1)$$

However, in case when domain $A$ is richer than $B$, the mapping $G : x \sim p_A \to y \sim p_B$ is many-to-one (i.e. if for one image $x \sim p_B$ there are multiple correct correspondences $y \sim p_A$), the generator is still forced to perfectly reconstruct the input even though some of the information of the input image is lost after the translation to the domain $B$. As shown in [5], such behavior of a CycleGAN can be described as an adversarial attack, and in fact, for any given image it is possible to generate such structured noise that would lead to reconstruction of the target image [5].

In practice, CycleGAN and other methods that utilize cycle-consistency loss add a very low-amplitude signal to the translation $\hat{y}$ that is invisible for a human eye. Addition of a certain signal is enough

to reconstruct the information of image $x$ that should not be present in $\hat{y}$. This makes methods that incorporate cycle-consistency loss sensitive to low-amplitude high-frequency noise since that noise can destroy the hidden signal (shown in Figure 3). In addition, such behavior can force the model to converge to a non-optimal solution or even diverge since by adding structured noise the model "cheats" to minimize the reconstruction loss instead of learning the correct mapping.

## 4 Defense techniques

### 4.1 Adversarial training with noise

One approach to defend the model from a self-adversarial attack is to train it to be resistant to the perturbation of nature similar to the one produced by the hidden embedding. Unfortunately, it is impoossible to separate the pure structured noise from the traslated image, so classic adversarial defense training cannot be used in this scenario. However, it is possible to prevent the model from learning to embed by adding perturbations to the translated image before reconstruction. The intuition behind this approach is that adding random noise of amplitude similar to the hidden signal disturbs the embedded message. This results in high reconstruction error, so the generator cannot rely on the embedding. The modified noisy cycle-consistency loss can be described as follows:

$$\mathcal{L}_{rec}^{noisy} = \|F(G(x) + \Delta(\theta_n)) - x\|_1 \,, \tag{2}$$

where $\Delta(\theta_n)$ is some high-frequency perturbation function with parameters $\theta_n$. In our experiments we used low-amplitude Gaussian noise with mean equal to zero. Such a simplistic defense approach is very similar to the one proposed in [33] where the discriminator is defended from the generator attack by regularizing the discriminator objective using the adversarial vectors. In our setting, however, the attack is targeted on both the discriminator and the generator of opposite domain, which makes it harder to find the exact adversarial vector. Which is why we regularize both the discriminator and generator using random noise. Since adding noise to the input image is equivalent to penalizing large magnitude of the gradients of the loss function, this also forces the model to learn smoother boundaries and prevents it from overfitting.

### 4.2 Guess Discriminator

Ideally, the self-adversarial attack should be detected by the discriminator, but this might be too hard for it since it never sees real and fake examples of the same content. In the supervised setting, this problem is naturally solved by conditioning the outputs on the ground truth labels. For example, a self-adversarial attack does not occur in Conditional GANs because the discriminator is conditioned on the ground truth class labels and is provided with real and fake examples of each class. In the unsupervised setting, however, there is no such information about the class labels, and the discriminator only receives unpaired real and fake examples from the domain. This task is significantly harder for the discriminator as it has to learn the distribution of the whole domain. One widely used defense strategy is adding the adversarial examples to the training set. While it is possible to model the adversarial attack of the generator, it is very time and memory consuming as it requires training an additional network that generates such examples at each step of training the GAN. However, we can use the fact that cycle-consistency loss forces the model to minimize the difference between the input and reconstructed images, so we can use the reconstruction output to provide the fake example for the real input image as an approximation of the adversarial example.

Thus, the defense during training can be formulated in terms of an additional *guess discriminator* that is very similar to the original GAN discriminator, but receives as input two images – input and reconstruction – in a random order, and "guesses" which of the images is fake. As with the original discriminator, the guess discriminator $D_{guess}$ is trained to minimize its error while the generator aims to produce such images that maximize it. The guess discriminator loss or *guess loss* can be described as:

$$\mathcal{L}_{guess} = \begin{cases} G_{guess}^A\{\mathbf{X}, F(G(\mathbf{X})\}, & \text{with probability 0.5} \\ 1 - G_{guess}^A\{F(G(\mathbf{X})), \mathbf{X}\}, & \text{with probability 0.5} \end{cases} \tag{3}$$

where $X \sim P_A$, $G_{guess}^A(\mathbf{X}, \hat{\mathbf{X}}) \in [0,1]$. This loss resembles the class label conditioning in the Conditional GAN in the sense that the guess discriminator receives real and fake examples that are presumably of the same content, therefore the embedding detection task is significantly simplified.

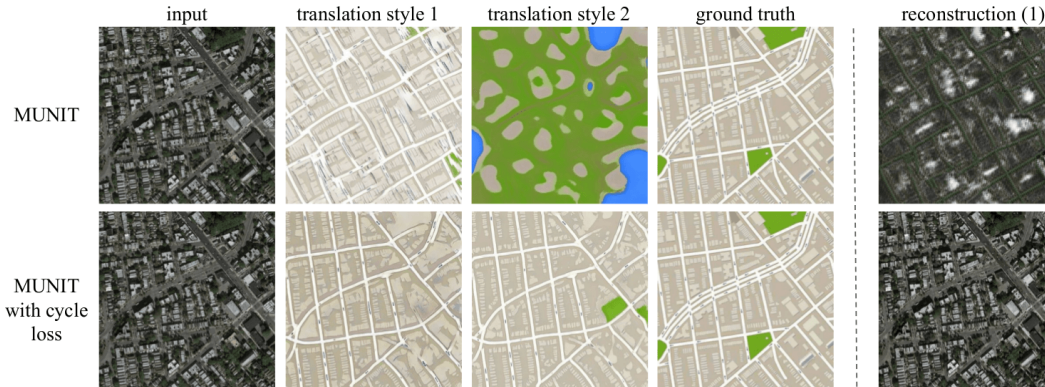

| | input | translation style 1 | translation style 2 | ground truth | reconstruction (1) |

Figure 2: Comparison of translation results produced by original MUNIT method and MUNIT with additional cycle-consistency loss. In columns 2 and 3 are shown the translation results with two different randomly generated style vectors. It can be observed that, while both methods incorrectly disentangled style and content information, the method that contains cycle-consistency loss forces the model to preserve the overall scene layout and produce more reliable translation in general. Column 5 shows the results of reconstruction of the input image from the maps with the first random style (column 2). More examples on Google Maps translation can be found in the supplementary material. *Best viewed in color.*

In addition to the defense approaches described above, it is beneficial to use the fact that the relationship between the domains is one-to-many. One naive solution to add such prior knowledge is by assigning a smaller weight to the reconstruction loss of the "richer" domain (e.g. photos in maps-to-photos experiment). Results of our experiments show substantial improvement in the generation quality when such a domain relation prior is used.

## 5 Experiments and results

In abundance of GAN-based methods for unsupervised image translation, we limited our analysis to three popular state-of-art models that cover both unimodal and multimodal translation cases: CycleGAN[34], UNIT[20] and MUNIT[11]. The details on model architectures and choice of hyperparameters used in our experiments can be found in the supplementary materials.

### 5.1 Datasets

To provide empirical evidence of our claims, we performed a sequence of experiments on three publicly available image-to-image translation datasets. Despite the fact that all three datasets are paired and hence the ground truth correspondence is known, the models that we used are not capable of using the ground-truth alignment by design and thus were trained in an unsupervised manner.

**Google Aerial Photo to Maps** dataset consisting of 3292 pairs of aerial photos and corresponding maps. In our experiments, we resized the images from $600 \times 600$ pixels to $400 \times 400$ pixels for MUNIT and UNIT and to $289 \times 289$ pixels for CycleGAN. During training, the images were randomly cropped to $360 \times 360$ for UNIT and MUNIT and $256 \times 256$ for CycleGAN. The dataset is available at [6]. We used 1098 images for training and 1096 images for testing.

**Playing for Data (GTA)**[26] dataset that consists of 24966 pairs of image frames and their semantic segmentation maps. We used a subset of 10000 frames (7500 images for training, 2500 images for testing) with day-time lighting resized to $192 \times 192$ pixels, and randomly cropped with window size $128 \times 128$. **SynAction** [28] synthetic human action dataset consisting of a set of 20 possible actions performed by 10 different human renders. For our experiments, we used two actors and all existing actions to perform the translation from one actor to another; all other conditions such as background, lighting, viewpoint etc. are chosen to be the same for both domains. We used this dataset to test whether the self-adversarial attack is present in the one-to-one setting. The original images were resized to $512 \times 512$ and cropped to $452 \times 452$. We split the data to 1561 images in each domain for training 357 images for testing.

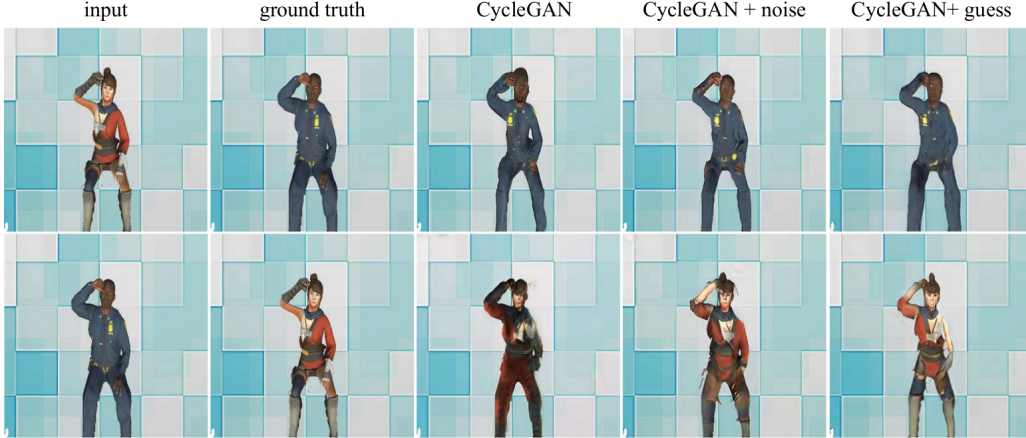

| input | ground truth | CycleGAN | CycleGAN + noise | CycleGAN+ guess |

Figure 3: SynAction actor translation example with CycleGAN, CycleGAN with noise and CycleGAN with guess loss.

## 5.2 Metrics

**Translation quality.** The choice of aligned datasets was dictated by the need to quantitatively evaluate the translation quality which is impossible when the ground truth correspondence is unknown. However, even having the ground truth pairs does not solve the issue of quality evaluation in one-to-many case, since for one input image there exist a large (possibly infinite) number of correct translations, so pixelwise comparison of the ground truth image and the output of the model does not provide a correct metric for the translation quality.

In order to overcome this issue, we adopted the idea behind the Inception Score [27] and trained the supervised Pix2pix[12] model to perform many-to-one mapping as an intermediate step in the evaluation. Considering the GTA dataset example, in order to evaluate the unsupervised mapping from segmentation maps to real frames (later on – segmentation to real), we train the Pix2pix model to translate from real to segmentation; then we feed it the output of the unsupervised model to perform "honest" reconstruction of the input segmentation map, and compute the Intersection over Union (IoU) and mean class-wise accuracy of the output of Pix2Pix when given a ground truth example and the output of the one-to-many translation model. For any ground truth pair $(A_i, B_i)$, the one-to-many translation quality is computed as $\text{IoU}(pix(G_A(B_i)), pix(A_i))$, where $pix(\cdot)$ is the translation with Pix2pix from $A$ to $B$. The "honest reconstruction" is compared with the Pix2pix translation of the ground truth image $A_i$ instead of the ground truth image itself in order to take into account the error produced by the Pix2pix translation.

**Reconstruction honesty.** Since it is impossible to acquire the structured noise produced as a result of a self-adversarial attack, there is no direct way to either detect the attack or measure the amount of information hidden in the embedding.

In order to evaluate the presence of a self-adversarial attack, we developed a metric that we call *quantized reconstruction honesty*. The intuition behind this metric is that, ideally, the reconstruction error of the image of the richer domain should be the same as the one-to-many translation error if given the same input image from the poorer domain. In order to measure whether the model is independent of the origin of the input image, we quantize the many-to-one translation results in such way that it only contains the colors from the domain-specific palette. In our experiments, we approximate the quantized maps by replacing the colors of each pixel by the closest one from the palette. We then feed those quantized images to the model to acquire the "honest" reconstruction error, and compare it with the reconstruction error without quantization. The honesty metric for a one-to-many reconstruction can be described as follows:

$$RH = \frac{1}{N} \sum_{i=1}^{N} \{\|G_A(\lfloor G_B(X_i) \rfloor) - Y_i\|_2 - \|G_A(G_B(X_i)) - Y_i\|_2\}, \tag{4}$$

where $\lfloor * \rfloor$ is a quantization operation, $G_B$ is a many-to-one mapping, $(X_i, Y_i)$ is a ground truth pair of examples from domains $A$ and $B$.

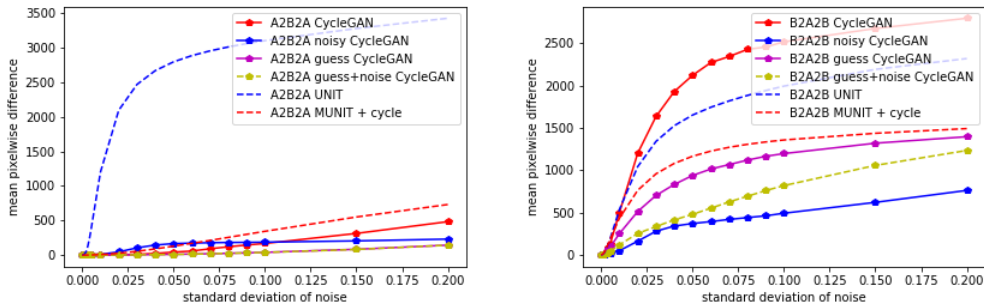

Figure 4: Illustration of sensitivity (Eq. 5) of cycle-consistent translation methods to high-frequency perturbations in one-to-many (left) and in many-to-one (right) cases. Here the domains A and B are segmentation maps and GTA video frames respectively. If the method is robust to the random perturbations then the reconstruction error should grow linearly with the amplitude of the added noise. For the cycle-consistent methods, we observe exponential growth of the reconstruction error in the one-to-many mapping that saturates at $\sigma = 0.09$, which means that these methods are highly sensitive to noise.

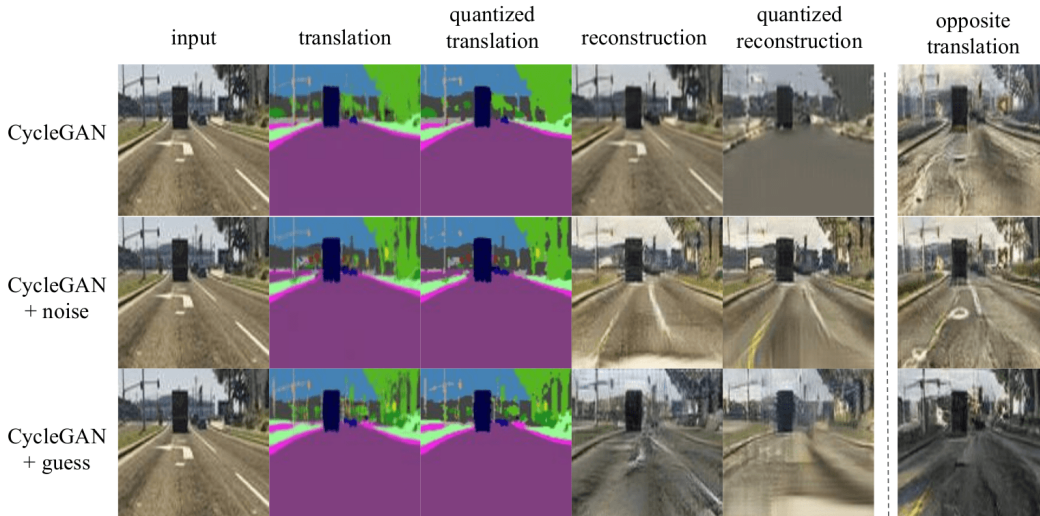

Figure 5: Quantized reconstruction results of the original CycleGAN, CycleGAN with noise defense and CycleGAN with guess loss defense. After translating the input GTA frame to the semantic translation map, we performed quantization such that the resulting translation would only contain the colors present in the real segmentation maps. We then fed the quantized translation results to reconstruct the input image (column 5). The last column represents the translation from the corresponding ground truth semantic segmentation map to real frame for comparison. As with random noise, quantization removes structured self-adversarial noise needed to accurately reconstruct the input, therefore the quantized reconstruction with CycleGAN differs drastically from the non-quantized reconstruction. CycleGAN with the guess loss and noisy CycleGAN, on the other hand, rely more on the input segmentation map than the original CycleGAN, therefore the quantized reconstruction is similar to the original reconstruction. More quantized translation examples can be found in the supplementary material. *Best viewed in color.*

**Sensitivity to noise.** Aside from the obvious consequences of the self-adversarial attack, such as convergence of the generator to a suboptimal solution, there is one more significant side effect of it – extreme sensitivity to perturbations. Figure 1 shows how addition of low-amplitude Gaussian noise effectively destroys the hidden embedding thus making a model that uses cycle-consistency loss unable to correctly reconstruct the input image. In order to estimate the sensitivity of the model, we add zero-mean Gaussian noise to the translation result before reconstruction and compute the reconstruction error. The sensitivity to noise of amplitude $\sigma$ for a set of images $X_i \sim p_A$ is computed

| Method | acc. segm ↑ | IoU segm↑ | IoU p2p↑ | RH↓ | SN↓ |
|---|---|---|---|---|---|
| CycleGAN | 0.23 | 0.16 | 0.20 | 27.43 ± 6.1 | 446.9 |
| CycleGAN + noise* | **0.24** | 0.17 | 0.23 | 9.17 ± 7.4 | **94.2** |
| CycleGAN + guess* | 0.24 | **0.17** | 0.21 | 11.4 ± 7.0 | 212.6 |
| CycleGAN + guess + noise* | 0.236 | **0.17** | **0.24** | **6.1 ± 5.9** | 150.6 |
| UNIT | 0.08 | 0.04 | 0.06 | 6.4 ± 11.7 | 361.5 |
| MUNIT + cycle | 0.13 | 0.08 | 0.17 | 2.5 ± 8.9 | 244.9 |
| pix2pix (supervised) | 0.4 | 0.34 | – | – | – |

Table 2: Results on the GTA V dataset. *acc. segm* and *IoU segm* represent mean class-wise segmentation accuracy and IoU, *IoU p2p* is the mean IoU of the pix2pix segmentation of the segmentation-to-frame mapping; *RH* (Eq.4) and *SN*(Eq.5) are the quantized reconstruction honesty and sensitivity to noise of the many-to-one mapping (B2A2B) respectively. * – our proposed defense methods. The reconstruction error distributions plots can be found in the supplementary material (Section 2).

| Method | acc. segm↑ | IoU segm↑ | IoU p2p↑ | RH ↓ | SN↓ |
|---|---|---|---|---|---|
| CycleGAN | 0.23 | 0.18 | 0.21 | 21.8 ± 5.2 | 251.2 |
| CycleGAN + noise* | 0.24 | 0.19 | 0.22 | 12.27 ± 4.42 | **222.2** |
| CycleGAN + guess* | 0.24 | 0.184 | **0.224** | 7.5 ± 2.4 | 235.4 |
| CycleGAN + guess + noise* | **0.25** | **0.19** | 0.22 | **-0.45 ± 2.3** | 238.3 |
| UNIT | 0.21 | 0.15 | 0.12 | 19.6 ± 6.1 | 528.2 |
| MUNIT + cycle | 0.15 | 0.09 | 0.12 | 21.4 ± 7.9 | 687.3 |
| pix2pix (supervised) | 0.3 | 0.23 | – | – | – |

Table 3: Results on the Google Maps dataset. The notation is same as in the Table 2.

by the following formula:

$$SN(\sigma) = \frac{1}{N} \sum_{i=1}^{N} \|G_A(G_B(X_i) + \mathcal{N}(0, \sigma)) - G_A(G_B(X_i))\|_2 \tag{5}$$

The overall sensitivity of a method is then computed as an area under curve of $AuC(SN(\sigma)) \approx \int_a^b SN(x)dx$. In our experiments we chose $a = 0$, $b = 0.2$, $N = 500$ for Google Maps and GTA experiments and $N = 100$ for the SynAction experiment. In case when there is no structured noise in the translation, the reconstruction error should be proportional to the amplitude of added noise, which is what we observe for the one-to-many mapping using MUNIT and CycleGAN. Surprisingly, UNIT translation is highly senstive to noise even in one-to-many case.

| Method | MSE↓ | SN ↓ |
|---|---|---|
| CycleGAN | 32.55 | 6.5 |
| CycleGAN+noise* | **22.18** | **1.1** |
| CycleGAN+guess* | 23.57 | 2.4 |
| CycleGAN+guess+noise* | 23.13 | 1.35 |

Table 1: Results on SynAction dataset: mean square error of the translation and sensitivity to noise.

The many-to-one mapping result (Figure 3), in contrast, suggests that the structured noise is present, since the reconstruction error increases rapidly and quickly saturates at noise amplitude 0.08. The results of one-to-many and many-to-one noisy reconstruction show that both noisy CycleGAN and guess loss defense approaches make the CycleGAN model more robust to high-frequency perturbations compared to the original CycleGAN.

## 5.3   Results

The results of our experiments show that the problem of self-adversarial attacks is present in all three cycle-consistent methods we examined. Surprisingly, the results on the SynAction dataset had shown that self-adversarial attack appears even if the learned mapping is one-to-one (Table 1). Both defense techniques proposed in Section 4 make CycleGAN more robust to random noise and increase its translation quality (see Tables 1, 2 and 3). The noise-regularization defense helps the CycleGAN

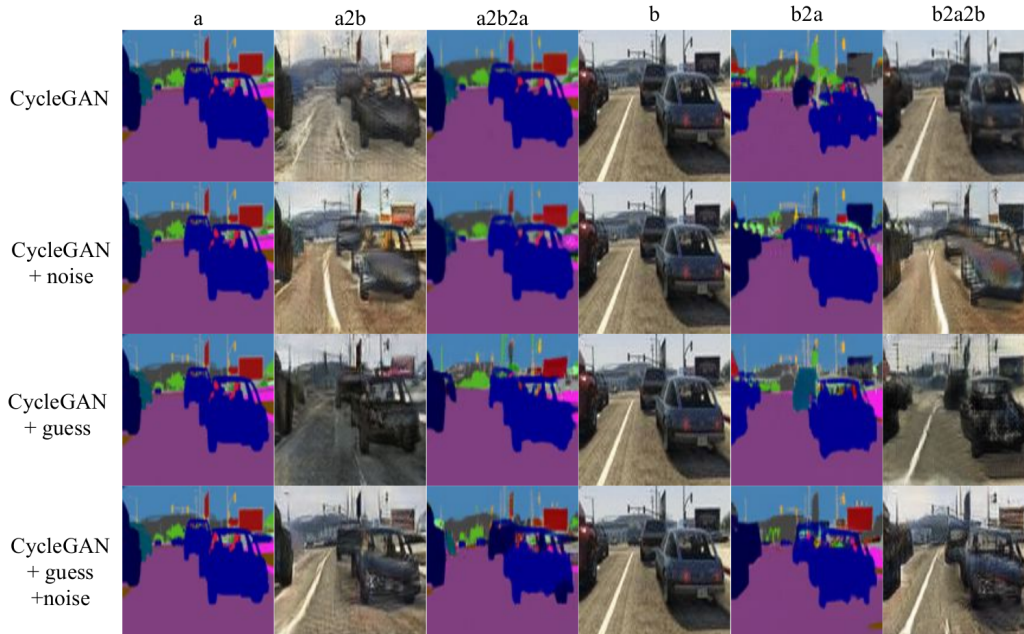

Figure 6: Results of the GTA frames-to-segmentation translation with the original CycleGAN and our defense techniques. The frame reconstruction (b2a2b) with noisy CycleGAN is remarkably similar to the opposite translation (a2b). For example, the road marking in the reconstructed image is located at the same place as in the translation (a2b) rather than as in the input (b).

model to become more robust both to small perturbations and to the self-adversarial attack. The guess loss approach, on the other hand, while allowing the model to hide some small portion of information about the input image (for example, road marking for the GTA experiment), produces more interpretable and reliable reconstructions. Furthermore, combination of both proposed defense techniques results beats both methods in terms of translation quality and reconstruction honesty (Figure 6).

Since both defense techniques force the generators to rely more on the input image than on the structured noise, their results are more interpretable and provide deeper understanding of the methods "reasoning". For example, since the training set did not contain any examples of a truck that is colored in white and green, at test time the guess-loss CycleGAN approximated the green part of the truck with the "vegetation" class color and the white part with the building class color (see Section 3 of the supplementary material); the reconstructed frame looked like a rough approximation of the truck despite the fact that the semantic segmentation map was wrong. This can give a hint about the limitations of the given training set.

## 6 Conclusion

In this paper, we introduced the self-adversarial attack phenomenon of unsupervised image-to-image translation methods – the hidden embedding performed by the model itself in order to reconstruct the input image with high precision. We empirically showed that self-adversarial attack appears in models when the cycle-consistency property is enforced and the target mapping is many-to-one. We provided the evaluation metrics that help to indicate the presence of self-adversarial attack, and a translation quality metric for one-to-many mappings. We also developed two adversarial defense techniques that significantly reduce the hidden embedding and force the model to produce more "honest" results, which, in return, increases its translation quality.

## 7 Acknowledgements

This project was supported in part by NSF and DARPA.

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
