[Supplementary Material · neurips_2019_supplementary.pdf]

# Adversarial Self-Defense for Cycle-Consistent GANs Supplementary Material

Dina Bashkirova [1], Ben Usman[1], and Kate Saenko [1,2]

[1]Boston University
[2]MIT-IBM Watson AI Lab
{dbash,usmn,saenko}@bu.edu

## 1   Model description and parameters

In our experiments, we used the implementation of **CycleGAN** provided at `https://github.com/junyanz/pytorch-CycleGAN-and-pix2pix`. For all CycleGAN models we used (original, noisy and guess-loss based) we set all the CycleGAN parameters to the default ones provided in the implementation except for the weights of the cycle-consistency loss. You can find the code of our project at `https://github.com/dbash/pix2pix_cyclegan_guess_noise`.

The CycleGAN parameters used in our experiments are:

- Generator architecture – ResNet with 9 residual block layers
- Discriminator architecture – 3-layer PatchGAN with patch size $70x70$ .
- Weight initialization – gaussian
- Instance normalization
- GAN objective – LSGAN
- Optimizer – Adam with momentum 0.5
- Learning rate – 0.0002 with linear policy
- Trained for 200 epochs.

The parameters specific to the proposed defense techniques are:

- For training with additive noise: standard deviation of noise $\sigma$ that should lie in the interval $[0, 1]$. The higher is the value of $\sigma$, the harder it is for the model to perform the self-adversarial attack. We chose the minimal value which results in the reconstruction that lacks the high-frequency details that should be lost after the translation, such as road texture or color.
- For the guess loss – weight of the guess loss $\lambda_{guess}$. We chose $\lambda_{guess}$ and the cycle-consistency losses weights $\lambda_A$ and $\lambda_B$ such that their corresponding loss values are of the similar magnitude during training. In other words, we choose the loss weights to be such that they all lie within one range and none of them dominates in the overall loss.
- For the guess loss + noise – weight of the guess loss $\lambda_{guess}$ and standard deviation of noise $\sigma$.

For the **GTA dataset**, the defense-specific parameters are:

- CycleGAN: $\lambda_A = 10, \lambda_B = 10$. We performed the experiments with on the CycleGAN with the smaller weights $\lambda_A$ and $\lambda_B$ that are proportional to the cross-domain relation as for the guess loss approach (e.g. $\lambda_A = 5$ and $\lambda_B = 3$), and this resulted in unreliable translation.

- CycleGAN + noise: $\sigma = 0.06$, $\lambda_A = 5$, $\lambda_B = 3$.
- CycleGAN + guess loss: $\lambda_{guess} = 2$, $\lambda_A = 1.5$, $\lambda_B = 1$.
- CycleGAN + guess loss + noise: $\lambda_{guess} = 2.5$, $\lambda_A = 2$, $\lambda_B = 1.5$, $\sigma = 0.03$.

For the **SynAction**, the defense-specific parameters are:

- CycleGAN: $\lambda_A = 10$, $\lambda_B = 10$.
- CycleGAN + noise: $\sigma = 0.1$, $\lambda_A = 10$, $\lambda_B = 10$.
- CycleGAN + guess loss: $\lambda_{guess} = 1$, $\lambda_A = 2$, $\lambda_B = 2$.
- CycleGAN + guess loss: $\lambda_{guess} = 2$, $\lambda_A = 25.$, $\lambda_B = 2.5$, $\sigma = 0.05$.

For the **Google Maps dataset**, we used the following parameters:

- CycleGAN: $\lambda_A = 10$, $\lambda_B = 10$.
- CycleGAN + noise: $\sigma = 0.06$, $\lambda_A = 10$, $\lambda_B = 10$.
- CycleGAN + guess loss: $\lambda_{guess} = 1$, $\lambda_A = 1$, $\lambda_B = 2$.
- CycleGAN + guess loss + noise: $\lambda_{guess} = 3$, $\lambda_A = 2$, $\lambda_B = 2.5$, $\sigma = 0.05$.

We based our experiments on the **UNIT and MUNIT** models on their original implementation: `https://github.com/NVlabs/MUNIT`.

UNIT architecture and parameters are:

- Optimizer – Adam with momentum 0.5 and second momentum 0.999
- Initialization – Kaiming
- Learning rate – 0.0001 with step decay policy (decay weight 0.5, step size 10000 iterations)
- weight on image reconstruction loss – 10
- weight on cycle-consistency loss – 10
- – weight of KL loss for cycle consistency – 0.01.
- Discriminator – 4-layer multiscale LSGAN with leaky ReLU activation function and 3 scales.
- Generator – VAE with ReLU activations, with 64 filters in the first layer, 2 downsampling layers and 4 residual blocks for the content encoder and decoder.
- Padding – reflect.

MUNIT parameters are:

- Optimizer – Adam with momentum 0.5 and second momentum 0.999
- Initialization – Kaiming
- Learning rate – 0.0001 with step decay policy (decay weight 0.5, step size 10000 iterations)
- weight on image reconstruction loss – 10
- weight on explicit cycle-consistency loss – 1
- – weight of KL loss for cycle consistency – 0.01.
- Discriminator – 4-layer multiscale LSGAN with leaky ReLU activation function and 3 scales.
- Generator – VAE with ReLU activations, with 64 filters in the first layer, with 256 filters in MLP, 2 downsampling layers and 4 residual blocks for the content encoder and decoder.
- Padding – reflect.
- Length of style code – 8

The code for the guess loss CycleGAN and noisy CycleGAN can be found in files "cycle_gan_guess_model.py" and "cycle_gan_noisy.py" respectively. In order to train or test the model, please add them to the folder "models" of the original CycleGAN project (`https://github.com/junyanz/pytorch-CycleGAN-and-pix2pix`) and specify the model parameter as "cycle_gan_guess" or "cycle_gan_noisy" instead of "cycle_gan".

Figure 1: GTA.**Left:** Difference in the error distribution of the non-quantized vs quantized reconstructions, **right:** Reconstruction Honesty distributions.

Figure 2: Google Maps. **Left:** Difference in the error distribution of the non-quantized vs quantized reconstructions, **right:** Reconstruction Honesty distributions.

## 2 Statistics

Figure 3: Sensitivity to noise on the Google Maps dataset. **Left:** translation from map to photo to map, **right:** translation from photo to map to photos.

Figure 4: Sensitivity to noise on the SynAction dataset. **Left:** translation from actor A to actor B, **right:** translation from actor B to actor A.

## 3 Translation Results Figures.

|  | a | a2b | a2b2a | b | b2a | b2a2b |
|---|---|---|---|---|---|---|

CycleGAN

CycleGAN + noise

CycleGAN + guess

Figure 5: Results of translation of GTA frames to semantic segmentation maps.

| input | translation | reconstruction |
|---|---|---|

Figure 6: Example of translation and reconstruction with CycleGAN + guess loss.

Figure 7: Noisy reconstruction.

Figure 8: Truck translation and reconstruction example with CycleGAN + guess loss.

Figure 9: Noisy reconstruction.

Figure 10: Quantized reconstruction of CycleGAN, UNIT and MUNIT.

Figure 11: Quantized reconstruction of CycleGAN, CycleGAN + noise and CycleGAN + guess loss.

Figure 12: Translation result with the proposed defense techniques.

Figure 13: Noisy reconstruction result.

Figure 14: Noisy reconstruction.

Figure 15: Results of translation of SynAction actors.