[Reviews · NeurIPS 2019]

Reviewer 1



Originality: The task is new to me. Adding noise for GAN training is not new. The guess discriminator seems new to me. Quality: The paper's claim is supported by its experiment results. I think this is a complete piece of work. Clarity: The paper is ok with clarity. I'd like to see a detailed model structure, especially for the guess discriminator. Significance: The paper proposed a few metrics to evaluate the quality of the model, which could be very useful for comparing different methods. Unfortunately, in this paper there were no other methods to compare with. So it is hard to say if their method is much better than existing methods.

Reviewer 2



The submission is clearly building upon the observations made in [5], and extends/complements them in meaningful ways. In particular, it contributes mitigation techniques as well as improved/complementary evaluation metrics. Overall, the submission is written clearly, and remains very readable in all parts. Although not strictly part of this evaluation, the provided supplementary material is exemplary, and can help reproducing these results. I see the submission as a high-quality contribution to 1) gain deeper insight into the workings of [unpaired] image-to-image translation systems, and 2) improve their quality. Both of these goals have been reached, by means of the contributions a)-c). The presented defense techniques in Section 4 are based more on empirical observation (i.e. results get better) than on provable guarantees, but this does not diminish their usefulness and level of significance. While the adversarial training with noise (Section 4.1) is a rather obvious approach (and even referred to by the authors as a "simplistic defense approach"), the guess discriminator loss in Section 4.2 is a more interesting modification. The loss terms are generic enough to be suitably applied to any kind of cyclic/reconstruction based image-to-image translation architecture. The experimental results are convincing, both in terms of the data sets they have been evaluated on, as well as in terms of the results. Experiments overall are thorough enough to be significant. It would have been even better to see what a combination of the two loss terms can achieve., i.e. another row "CycleGAN + noise* + guess*" in Tables 2 and 3 (after optimization of the loss weighting hyperparameters). The novel "metrics" are quite ad-hoc but make sense, and appear to provide further insight into the behavior of these GAN-based translation networks. Coming up with good metrics here is not that easy, so this contribution is appreciated. The sensitivity-to-noise metric should be directly improved by the noise defense, and, unsurprisingly, this approach yields the best results under the metric. Minor comments: - References [10] and [11] are the same paper. - I think there is some word missing in the sentence starting in l. 102. I get the meaning, though. - Extraneous word ('is') in l. 112 - Typo in l. 156: 'Coditional' -> 'Conditional'

Reviewer 3



I think the self-adversarial attack observation is quite interesting but not very convinced that the proposed defense techniques are novel enough for the submission. Note self-adversarial attack is not a new observation(as the paper heavily cited), and both defense techniques (adding noise and adding pairwise discriminator) exist in the literature. Pros: This paper is quite well written and properly summarized the related works. This paper shows significant effort in conducting experiments. Cons: Novelty is not enough as most of the proposed solution or observations are already published. Need more insight on the proposed solutions instead of similar to some other works.

[Author Response · NeurIPS 2019]

We cannot thank the reviewers enough for their valuable feedback on our work. Below we provide the response and
comments on their remarks and questions.

**Reviewers 1 and 2: Combine guess loss with additive noise.** Due to the time constraints of the rebuttal, we limited
ourselves to a single setup: two methods combined with the same sets hyperparameters as in the main paper. This
combination of the guess loss with the additive noise beats the out-of-the-box CycleGAN on the GTA dataset in terms
of the translation accuracy but performed weaker than the individual solutions we proposed, supposedly due to the
non-optimal choice of hyperparameters (weight of the guess loss and $\sigma$ of the Gaussian noise). We will test more
hyperparameters and provide an extended analysis for all three metrics in the camera-ready version of the paper.

**Reviewer 1: No other methods to compare with, so it is hard to say if their method is much better than existing**
**methods.** To our knowledge, we are the first to develop a defense technique that addresses specifically the self-
adversarial attack. Most recent advances in adversarial defense methods address "black-box attacks" performed by a
third party against a fixed model using additive signal with known properties (*i.e.* bounded norm) that alters models
predictions in a specified way. In the self-adversarial setting, however, the attack is performed by the generative model
itself to reconstruct the information that is lost during translation and is a natural consequence of the imposed cycle
constraint. Since the self-adversarial attack is performed implicitly during the translation, we can not extract the
embedded signal, or even understand its true nature or measure its properties. Also, the "attacker" in this case constantly
adapts to the setting and fine-tunes the embedding as the discriminator learns to detect it. Therefore, black-box methods
are of lesser use for the self-adversarial defense. Moreover, both the additive noise and the guess loss methods build
upon ideas of state-of-art defenses against the white-box adaptive attacks, namely, gradient penalties and the "adversarial
training". The latter incorporates adversarial examples during training to increase the model's robustness to the attack.
Since we cannot explicitly model the structured noise produced by the self-adversarial attack, and cannot acquire the
non-adversarial translations that do not contain the self-adversarial noise, we cannot apply the adversarial training
directly to each of the two translation networks. Instead, we note that the reconstructed image tends to be almost
identical to the input but must contain the adversarial noise since the model is not aware of the origin of the input.
Therefore the reconstructed image can serve as an adversarially perturbed example of the non-adversarial input image.
We provide both non-adversarial input image and the adversarial reconstruction to the guess discriminator so that it
could detect and penalize the presence of the structured noise. Additionally, our goal was to improve the performance
of the cycle-consistent translation methods by defending them against the self-adversarial attack and thus making them
rely more on the visual characteristics of the input rather than on the hidden embedding, so we believe that comparing
our "defended" CycleGAN with the classic CycleGAN, UNIT, and MUNIT is a good baseline comparison.

**Reviewer 3: Novelty is not enough as most of the proposed solution or observations are already published.** While
the presence of the self-adversarial attack in the CycleGAN model was previously reported [5], we 1) show that this
phenomenon is present in all major unsupervised translation methods that incorporate the cycle-consistency loss; 2)
more importantly, we are the first to propose defense techniques against this particular attack, as well as 3) a set of
metrics that reveal the degree of embedding and the robustness of the model to the self-adversarial behaviour. While
adding noise is a heavily used technique (e.g. for regularization), we would like to stress that this paper is the first
systematic analysis of the effect the additive noise has on the robustness of the cyclic translation models against the
self-adversarial attack. As for the pairwise discriminator, we would like to emphasize that our loss discriminates an
image from its own perturbed version. That sets it aside from other pairwise GAN losses, such as the relativistic GAN
loss that predicts which of two *different* images is real and which is fake, conditional discriminators that use an image
together with the corresponding conditioner from a different domain (e.g. a segmentation map), and, to our knowledge,
all other actively used discriminator losses with multiple inputs. Moreover, no prior work utilized and evaluated the
effectiveness of such discriminators in defending GANs against adversarial attacks.

**Reviewer 3: I would suggest authors make more effort to justify the proposed defense techniques and providing**
**insight that why the defense techniques could help to solve the problem. E.g. how do I know if or not this**
**method actually forces the model to "hide" info in another way?**

Figure 5 in the original submission illustrates a qualitative method for determining whether a given model exhibits the
embedding behavior of any kind. Consider images with accurately estimated segmentation maps (A2B matches ground
truth B). We observe that the CycleGAN model produced perfect reconstructions (A2B2A) that are very different
from respective translations of ground truth segmentation maps (B2A), whereas reconstructions generated by the
models with either additive noise or the guess loss match respective segmentation translations much better, suggesting
that these models did not rely on any hidden information during reconstruction. More examples can be found in
the supplementary. Unfortunately, this intuitive qualitative metric is difficult to measure quantitatively as it requires
common sense understanding of features that could and could not be inferred from segmentation maps alone (e.g. road
marking position can be, but car colors can not), and the difficulty of estimating perceptual similarity between images of
natural scenes; the proposed "honesty" metric leverages a pre-trained pix2pix model to measure perceptual similarity.

[Meta-Review · NeurIPS 2019]

This paper addresses a problem in cycleGAN that was publicized in [5]. Two heuristic approaches are proposed, and shown to be improving on the sensitivity to noise. I agree with the reviewers on the decision to accept.